# Deep Learning Approaches to Osteosarcoma Diagnosis and Classification: A Comparative Methodological Approach

**DOI:** 10.3390/cancers15082290

**Published:** 2023-04-13

**Authors:** Ioannis A. Vezakis, George I. Lambrou, George K. Matsopoulos

**Affiliations:** 1Biomedical Engineering Laboratory, School of Electrical & Computer Engineering, National Technical University of Athens, 9 Iroon Polytechniou St., 15780 Athens, Greece; ivezakis@biomed.ntua.gr (I.A.V.); glamprou@med.uoa.gr (G.I.L.); 2Choremeio Research Laboratory, First Department of Pediatrics, National and Kapodistrian University of Athens, Thivon & Levadeias 8, 11527 Athens, Greece; 3University Research Institute of Maternal and Child Health & Precision Medicine, National and Kapodistrian University of Athens, Thivon & Levadeias 8, 11527 Athens, Greece

**Keywords:** osteosarcoma, neural networks, machine learning, deep learning

## Abstract

**Simple Summary:**

Osteosarcoma is a rare form of bone cancer that primarily affects children and adolescents during their growth years. Known to be one of the most aggressive tumors, its 5-year survival rate ranges from 27% to 65% across all age groups. Despite the availability of treatment options such as surgery, chemotherapy, and limb-salvage surgery, the risk of recurrence and metastasis remains high even after remission. To improve disease prognosis, it is crucial to explore new diagnostic and treatment methods. Machine learning and artificial intelligence hold promise in this regard. In this study, we adopted a comparative methodological approach to evaluate various deep learning networks for disease diagnosis and classification, aiming to contribute to the advancement of these promising technologies in the field of osteosarcoma research.

**Abstract:**

Background: Osteosarcoma is the most common primary malignancy of the bone, being most prevalent in childhood and adolescence. Despite recent progress in diagnostic methods, histopathology remains the gold standard for disease staging and therapy decisions. Machine learning and deep learning methods have shown potential for evaluating and classifying histopathological cross-sections. Methods: This study used publicly available images of osteosarcoma cross-sections to analyze and compare the performance of state-of-the-art deep neural networks for histopathological evaluation of osteosarcomas. Results: The classification performance did not necessarily improve when using larger networks on our dataset. In fact, the smallest network combined with the smallest image input size achieved the best overall performance. When trained using 5-fold cross-validation, the MobileNetV2 network achieved 91% overall accuracy. Conclusions: The present study highlights the importance of careful selection of network and input image size. Our results indicate that a larger number of parameters is not always better, and the best results can be achieved on smaller and more efficient networks. The identification of an optimal network and training configuration could greatly improve the accuracy of osteosarcoma diagnoses and ultimately lead to better disease outcomes for patients.

## 1. Introduction

Osteosarcoma (OS), or osteogenic sarcoma, is a malignancy of the bone derived from cells of mesenchymatic origin that exhibit osteoblastic differentiation [1,2]. It is a very aggressive type of tumor and is known to be the most common primary bone cancer. In OS, mesenchymal cells produce osteoid and immature bone tissue, primarily at the loci of bone growth. The immature bone tissue is known to be linked to osteoblast proliferation, and thus it is probable that cells acquire genomic aberrations as well as epigenetic changes.

OS is a disease with a high degree of heterogeneity, which partly explains the difficulty in understanding the molecular machinery underlying its pathogenesis. Several studies have investigated the molecular pathogenetic mechanisms of OS.

Proposed key players in OS pathology include the genes (or proteins) *Rb* [3], *TP53* [4], *Grim-19* [5,6], *P21*/*RAS* [7,8], and *NF-κB* [9]. *Rb*, a tumor suppressor gene, is aberrantly expressed in several tumors. This gene is known for its role in cell cycle progression and regulation of G1-to-S phase transition. The Rb protein is able to inhibit excessive cell growth, and upon phosphorylation, the pRb inhibits cell proliferation. One of the main findings concerning OS is that both the Rb and p53 (*TP53*) proteins are dysfunctional [10]. TP53, or tumor protein 53 is a “transcription factor that regulates critical genes in DNA damage response, cell cycle progression, and apoptosis pathways” [4]. TP53 acts as a tumor suppressor in all tumor types. In normal cells, the TP53 levels are low [11], while in tumors, TP53 is either mutated or down-regulated [11]. The major regulator of TP53 is MDM2, which acts as a negative regulator and can trigger the degradation of the p53-ubiquitin complex [12]. TP53 germline mutations are linked to early childhood OS. Another tumor suppressor gene known for its role in OS is *GRIM-19*. Its main function is to mediate apoptosis. Recent studies have indicated that GRIM-19 is downregulated in OS, while radiation-induced apoptosis in OS is tightly linked to TP53 upregulation, whose down-regulation infers resistance to radiation-induced apoptosis [5].

Another interesting key molecule in the biology of OS is the transcription factor NF-κB [13]. This is one of the well-studied transcription factors for its role in inflammation [14], tumor progression [15], chemotherapy (and radiation) resistance [16], and in particular for its role in OS radiation-related resistance [9,17,18]. Recent studies have shown that NF-κB activation is equivalent to tumor resistance in both chemotherapy as well as radiation therapy [17,18]. The mechanism of resistance to chemo- and radiation therapy in OS is still largely unknown, yet it has been reported that NF-κB functions through the Akt/NF-κB pathway [19,20]. In a very recent report, an explanation was given for this effect, which included evidence that osteosarcoma resistance comes about due to BMI-1 overactivation.

OS is the most prevalent primary skeletal malignancy of childhood and adolescence. It primarily occurs during the adolescent growth spurt between the ages of 10 and 14, and it accounts for 2% of all childhood neoplasms [21]. OS is considered to be a devastating disease. Although in recent years therapeutic advances and options have greatly improved patient survival, metastasis remains the main obstacle in patient prognosis [1]. Patient five-year survival has reached 60-65% in recent years, but the overall prognosis remains poor [1]. Unfortunately, in metastatic cases of the disease, overall survival is as low as 27% [22,23].

Although it is considered a rare form of cancer, approximately 1000 children are newly diagnosed with OS each year in the US alone. In the past, amputations used to be the first line of treatment, aiming to remove the tumor completely. However, advances in imaging techniques, as well as neoadjuvant (preoperative) chemotherapy (NPC) and adjuvant (postoperative) chemotherapy (APC), have facilitated the shift to limb-salvage surgery and increased the five-year survival rate from <20% in the 1970s to 65% in 2015 [24,25]. Following NPC, the achievement of high tumor necrosis rates (>90%) is associated with a significantly higher survival rate and better prognosis [25,26]. Despite recent progress in diagnostic methods, both molecular and imaging histopathological assessment remains the current gold standard for treatment evaluation.

Histopathology includes the evaluation of tissue samples using microscopic examination. In the case of OS, microscopic examination facilitates the estimation of tumor differentiation, invasion, and the presence of necrosis. Still, microscopy is a lengthy, tedious process that is prone to observer bias [27,28]. OS’s high degree of heterogeneity further complicates this process. As such, automating the histopathological evaluation of OS could result in more accurate, fast, and cost-effective examinations [28].

Machine learning approaches are the current state-of-the-art (SotA) method for image classification. Conventional machine learning algorithms such as Support Vector Machines (SVMs) [29,30] and Random Forests (RF) [31,32,33,34] have been widely used in the past for image classification tasks. They rely on a set of features extracted from the images, and their performance is tightly coupled to feature quality. The feature extraction step itself is no easy task; it usually includes hand-crafted methods that fall short on large datasets with a high degree of variability. On the other hand, deep learning architectures such as Convolutional Neural Networks (CNNs) and Vision Transformers (ViTs) have achieved impressive results, comparable to human performance in many tasks. Numerous recent reports have highlighted the importance and advantages of deep learning over conventional machine learning approaches in microscopy [35,36,37]. Unlike SVMs and RFs, these methods do not depend on a feature extraction step; instead, they learn to perform the feature extraction on their own. Their success is highly dependent on the quality and size of the dataset used for training, as well as the overall design of the network architecture. For this reason, new network architectures emerge every year.

The use of machine learning in histopathology has a relatively short history, with the earliest reported studies dating back to the 1990s. These early studies primarily focused on the use of simple image analysis techniques, such as thresholding and edge detection, to identify specific structures within the images [38]. As technology progressed, researchers started exploring more sophisticated methods, such as texture analysis and pattern recognition, to improve the classification of Whole Slide Images (WSIs). WSIs are high-resolution digital images of stained tissue samples captured by a digital slide scanner. The scanner-produced images are large, typically several gigabytes in size and containing millions of pixels. Analyzing an image of this size poses a significant challenge, as current hardware capabilities are insufficient to process potentially thousands of images of this size.

In the present study, we trained several state-of-the-art networks using the same dataset and compared their results to determine which architecture, depth, and input image size was most effective in detecting viable and necrotic tumors, as well as healthy tissue.

## 2. Methodology

### 2.1. Methodological Description of Deep Learning Methodologies

The current state-of-the-art CNNs have been designed to work with images from ImageNet or similar databases, with an average resolution of 469 × 387 pixels [39,40]. In practice, these images are cropped or resized to 224 × 224 or 256 × 256 pixels to conserve memory and improve computational efficiency [41]. However, these images are much smaller than WSIs, which can contain up to 100,000 × 100,000 pixels [42]. To overcome current technology limitations, researchers typically analyze local mini-patches cropped from the WSIs, and each patch is classified independently. With increasing computational power and memory, larger patch sizes are becoming possible. In theory, a larger patch size should produce more accurate results, as it incorporates a much larger image context and more data points (pixels) [36]. Yet, in the present case, processing large images posed several challenges due to limitations in memory and processing power capacities.

### 2.2. Dataset

The dataset used in this study was the publicly available Osteosarcoma data from UT Southwestern/UT Dallas for Viable and Necrotic Tumor Assessment (https://wiki.cancerimagingarchive.net/pages/viewpage.action?pageId=52756935, accessed 31 January 2023 [43]) [44,45,46,47]. The patients included in the dataset were treated at the Children’s Medical Center in Dallas between 1995 and 2015, where they underwent surgical resection. The resected bone was then cut into pieces, de-calcified, treated with an H&E stain, and converted to slides [45]. The slides were scanned into digital WSIs, 40 of which were manually selected by two pathologists based on tumor heterogeneity and response characteristics. From each of these WSIs, 30 tiles of size 1024 × 1024 pixels were randomly selected, resulting in 1200 tiles. After filtering out non-tissue, ink-mark regions, and blurry images, 1144 tiles were selected for analysis. Each of these tiles was manually classified by the pathologists as Non-Tumor (NT), Viable Tumor (VT), or Necrosis (NC), with the following distributions: 536 (47%) NT, 263 (23%) NC, and 345 (30%) VT. It should be noted that 53 out of the 263 (20%) NC images also contained segments of VT. An example of the images is shown in Figure 1.

### 2.3. Experimental Setup

We compared the performance of various state-of-the-art deep learning architectures on the Osteosarcoma dataset. To train our models, we utilized transfer learning by fine-tuning networks that had been pre-trained on the ImageNet dataset. This allowed us to leverage the knowledge learned by the pre-trained model and apply it, with adjustments, to the much smaller Osteosarcoma dataset.

All computations were performed with the PyTorch framework. The networks were trained on a single NVIDIA Titan Xp GPU with 12GB of memory. The source code is available on Zenodo (https://doi.org/10.5281/zenodo.7765031, accessed 23 March 2023).

The chosen deep learning architectures included the Visual Geometry Group network (VGG) [48], the Residual Network (ResNet) [49], the MobileNetV2 [50], the EfficientNet family of networks [51], and the Vision Transformer [52]. We chose these architectures because they are well-established with proven success in image classification, as they have all previously achieved state-of-the-art results on ImageNet.

A particular network architecture can usually be scaled up or down in terms of size (i.e., number of parameters) depending on the requirements of a specific use case. While larger networks have greater learning capacities, they are also more prone to overfitting, particularly on small datasets. Therefore, in our study, we selected several variants of each network architecture, ranging from small (~2.2M parameters) to large (~86M parameters) models. Table 1 shows the number of parameters for each chosen network variant, which may differ from those reported in their original publications (ResNet [49], VGG [48], MobileNet [50], EfficientNet [51]) due to modifications made to their last layers. For example, a VGG16 normally has 138M parameters. However, in this study, we followed the work of Anisuzzaman et al. (2021) [53], where they substituted the last fully connected layer of size 4096 neurons (which makes up a classifier containing ~120M parameters), with two fully connected layers of sizes 512 and 1024 neurons, containing only ~13M parameters. This modification resulted in a much smaller network without significant changes to its architecture. Modifying the last layer of the networks was necessary because they were originally designed to classify images among 1000 categories, whereas in our use case, we only required classification among three categories.

To compare the performance of the different deep learning architectures, we split the dataset into a training and a test set, with a 70/30 split. We ensured that each network was trained and evaluated on the same set of images by using the same split each time, thus providing a fair comparison.

For all experiments, we used the Adam optimizer [54] with a decoupled weight decay [55] and a learning rate (LR) of 0.0003. We used a cosine annealing learning rate to reduce the LR from 3 × 10^−4^ to 1 × 10^−5^ over the 100 epochs of training. Although Adam may not have been the best optimizer for all tasks, it is a widely used default choice in the literature [46,53,54,56,57]. We choose to use it because our main goal was to compare the performance of different architectures rather than to optimize hyperparameters.

The batch size varied depending on the network architecture and the size of the input images. For the largest image size of 1024 × 1024 pixels, we set the batch size to 2, as larger batch sizes could not fit in the GPU memory for most networks. For networks where even a batch size of 2 did not fit into the GPU memory, we trained them on a smaller image size of 896 × 896 pixels instead. A notable exception was EfficientNetB7, which was too large to fit in memory even with the reduced image size.

In addition to training on the full-resolution images, we also trained the networks on down-sampled versions of the images, with resolutions of 512 × 512 and 256 × 256 pixels. We performed down-sampling using bilinear interpolation and doubled the batch size when the image size was halved. Down-sampling is frequently performed when the goal is image classification as opposed to segmentation, to reduce the computational cost and memory requirements while still achieving good results. Although down-sampling is a destructive process that removes information, it is not a major problem for image classification as the network is only interested in the class of the image, not the exact location of an object. Moreover, down-sampling can even be beneficial as it effectively enlarges the receptive field of the CNN’s convolutional layers [16], allowing the network to learn more global features. On the other hand, there is a trade-off between the benefits of the enlarged receptive field and the loss of information due to down-sampling. Therefore, it is not always clear which input image size is optimal.

Further to image resizing, we normalized each RGB channel of the input image independently by subtracting the mean and dividing by the standard deviation. To match the input images with those the networks were pre-trained with, we used the means and standard deviations of the ImageNet dataset, rather than the OS. The mean values used were 0.485, 0.456, and 0.406, and the standard deviation values were 0.229, 0.224, and 0.225 for the R, G, and B channels, respectively.

In addition, data augmentation techniques were applied during training to increase the diversity of the training set. Specifically, we used random horizontal and vertical flipping, as well as random rotation within 20 degrees.

### 2.4. Network Evaluation

We trained the following network architectures on the Osteosarcoma dataset: EfficientNetB0, EfficientNetB1, EfficientNetB3, EfficientNetB5, EfficientNetB7, MobileNetV2, ResNet18, ResNet34, ResNet50, VGG16, VGG19, and ViT-B/16. The ViT’s architecture was designed to treat the input image as a sequence of 14 patches with a size of 16 × 16 pixels, thus limiting the image size to exactly 224 × 224 pixels. As a result, we could not experiment with different image sizes for this model due to the need to load pre-trained weights.

We evaluated the performance of each trained network on the test set and computed the *F*1 score using the One-vs-Rest (OvR) multiclass strategy. More specifically:(1)$$F1=2\frac{precision\cdot recall}{precision+recall}$$
(2)$$precision=\frac{TP}{TP+FP}$$
(3)$$recall=\frac{TP}{TP+FN}$$
where *TP*, *TN*, *FP*, and *FN* are the true positives, true negatives, false positives, and false negatives, respectively. According to the OvR strategy, each class is treated as a binary classification problem, with the positive class being the class of interest and the negative class comprising all other classes. Therefore, the terms *TP* and *FP* refer to the number of images that were classified as belonging to the class of interest, with *TP* being the number of correct classifications and *FP* being the number of incorrect classifications. Similarly, *TN* and *FN* refer to the number of images that were classified as not belonging to the class of interest, with *TN* being the number of correct classifications and *FN* being the number of incorrect classifications.

### 2.5. Follow-Up Experiment

Following network evaluation and result analysis, we proceeded to select the best-performing combination of network and image-input size, and retrained it using five-fold cross-validation. This was done in order to provide a more accurate estimation of the model’s performance, which was not tied to a specific training-validation split. Specifically, we split the dataset into five parts (folds), using four of them to train the network and the remaining one to validate it. We repeated the process five times, with each fold serving as the validation set once. The final performance was then computed as the average over the five folds. The details of the retraining process remained the same as before, i.e., we used the same number of epochs, optimizer, learning rate, and augmentation strategy.

In this experiment, we computed additional performance metrics in order to conduct a more thorough investigation into the network’s performance and interpret its usefulness in a clinical setting. To this end, we computed the mean and standard deviation of the F1 Score, Accuracy, Specificity, Recall, and Precision across the five folds, as well as the combination of the Confusion Matrices through summation, and the Receiver Operating Characteristic (ROC) curve.

Expanding on the previous equations, the accuracy (Equation (4)) and specificity (Equation (5)) of the classifier were calculated as:(4)$$accuracy=\frac{TP+TN}{TP+FP+FN+TN}$$
(5)$$specificity=\frac{TN}{TN+FP}$$

The confusion matrix is a table that summarizes the performance of a classifier by comparing its predicted labels with the true labels of a test dataset. It provides a breakdown of the number of *TP*, *TN*, *FP*, and *FN* for each class, allowing for a more detailed evaluation of a model’s performance.

The Receiver Operating Characteristic (ROC) curve is a graphical representation of the performance of a binary classifier as its discrimination threshold is varied. It plots the true positive rate (recall) against the false positive rate for different threshold values. The Area under the ROC curve (AUC) is a commonly used metric to evaluate the overall performance of the classifier, with a higher AUC indicating better performance.

## 3. Results

### 3.1. Network Comparison

We compared the performance of the different architectures using the *F*1 score, which is a robust measure of classification performance considering both precision and recall. Precision measures the proportion of true positive predictions out of all positive predictions made by the model (Equation (2)), while recall measures the proportion of true positive predictions out of all actual positive samples (Equation (3)). The *F*1 score is calculated as the harmonic mean of the two values (Equation (1)), requiring both to contribute evenly in order to get a high *F*1 score value. Table 2 shows the *F*1 scores of the networks, providing a comparative analysis of their performance in each of the three classes (NT, VT, and NC). In order to compare the overall efficacy of the applied methodologies, we estimated the macro-average *F*1 score, which is presented in Appendix A. 

We found that deeper architectures do not necessarily perform better than shallower ones (Table 2, Appendix A). MobileNetV2, by far the smallest network in our experiments, yielded the highest macro-averaged *F*1 score. EfficientNetB0 performed equally or better than its variants B1, B3, B5, and B7. Furthermore, ResNet34 outperformed its bigger counterpart, ResNet50. This behavior is not surprising, given that we observed overfitting with almost every architecture where the accuracy on the training set approached 100% (Appendix A). Notable exceptions were the VGG networks, which failed to learn altogether except for VGG16, which achieved good results only on image input size of 256 × 256 pixels. This is due to the initial learning rate of 3 × 10^−4^ for the Adam optimizer being too large, causing large updates to the networks’ weights and converging on a suboptimal solution where every sample was classified as NT. Later experiments with the learning rate set to 1 × 10^−5^ provided results comparable to similarly sized networks. We chose not to include them in this study because (a) they did not impact our conclusions, and (b) we did not perform hyper-parameter optimization for other networks.

Increasing the image size did not appear to provide significant benefits, as it introduced new data points that could cause the model to overfit. This trend was observed in networks (EfficientNetB0, EfficientNetB1, ResNet18, ResNet34, ResNet50, and MobileNetV2). Some networks did achieve slightly higher results when trained on a larger input image size, including EfficientNetB3, EfficientNetB5, and EfficientNetB7. This small increase in performance could be attributed to EfficientNets utilizing compound scaling, a technique which uniformly scales the network’s width, depth, and input resolution between variants. Consequently, bigger EfficientNets were pre-trained on larger input image sizes, which could give them a slight advantage when fine-tuning on larger images.

### 3.2. Follow-Up Experiment

Based on the macro-averaged *F*1 score, we selected the MobileNetV2 with an input image size of 256 × 256 pixels as the best-performing configuration. The network was re-trained (as described in Section 2.5), and the mean ± standard deviation of the *F*1 score, accuracy, specificity, recall, and precision are summarized in Table 3. To provide a comprehensive overview of the classification performance, we aggregated the confusion matrices obtained from all folds by summation and presented the resulting confusion matrix containing all 1144 samples in Table 4.

The MobileNetV2 architecture was re-trained using 5-fold cross-validation to confirm the consistency of the results. The obtained mean *F*1 scores were 0.95 and 0.90 for NT and VT, respectively, indicating that the model performed well in identifying these categories. However, the *F*1 score for NC was lower, with a value of 0.85.

The precision values for NC and VT were similar, indicating that the model was equally confident when predicting either class. However, the lower recall value observed for NC suggests that the model had difficulty distinguishing this category from the other classes. Upon examination of the confusion matrix presented in Table 4, we found that out of 316 NC images, 24 were classified as NT and 30 were classified as VT. We also observed that when NT and VT were misclassified, they were seldom mistaken for each other but rather labeled as NC. This suggests that the observed lower accuracy for NC was not due to class imbalance within the dataset, as the network would have been biased towards classifying more samples as NT if that were the case. Instead, these findings suggest that NC shares visual features with both NT and VT, leading to misclassifications by the network. This behavior was observed throughout all of our experiments (Table 2). Indeed, as explained in Section 2.2, approximately 20% of the NC images also depicted VT. Re-training the network without these ambiguous images significantly increased the network’s accuracy (Table 5).

In order to provide an overview of MobileNetV2’s performance on each fold, we used ROC analysis and plotted the results in Figure 2. The different curves computed on each fold were superimposed in order to assist in performance comparison. The performance across all folds was consistent, with very high AUC values ranging from 0.98 to 1 for NT (Figure 2A), 0.97 to 0.99 for VT (Figure 2B), and 0.95 to 0.97 for NC (Figure 2C). The observed similarities across all AUCs for all classes indicates that the selected network is likely to achieve similar classification performance on different but comparable datasets.

## 4. Discussion

In the present study, we investigated various network architectures in order to examine their use in OS microscopy image classification.

### 4.1. Comparing Neural Networks

In order to obtain a better understanding of network efficiencies, we compared the results of MobileNetV2’s re-training with other studies that used the same osteosarcoma dataset (Table 6).

To ensure a fair comparison, we excluded studies that further modified the images using methods such as cropping patches to artificially enlarge the dataset. Such modifications have been shown before to improve performance [35] compared to uncropped tiles, but can also lead to data leakage if not done correctly. For example, two patches from the same tile located next to each other could end up in the training and test set. Consequently, the network would be evaluated on images that are extremely similar to the training set, resulting in accuracies that are too optimistic [58].

**Table 6 cancers-15-02290-t006:** Comparison of overall accuracy with related work in the same dataset.

Study	Method	Validation Strategy	Overall Accuracy
Arunachalam et al. (2019) [35]	Custom CNN	Holdout	0.910
Anisuzzaman et al. (2021) [53]	VGG19	Holdout	0.940
Bansal et al. (2022) [59]	Combination of HC and DL features	Holdout	0.995
Present study	MobileNetV2	Cross-Validation	0.910

Arunachalam et al. (2019) [35] developed a custom CNN architecture to classify the same osteosarcoma dataset. They reported their results on a limited subset of 230 images, as they performed an 80-20 split between the training and test set (holdout validation strategy). They achieved an overall accuracy of 0.91 and a recall of 89.5, 92.6, and 91.5 for NT, VT, and NC, respectively. Although the authors did not report specific implementation details of the re-training procedure, such as the image input size when using full-size image tiles, a key difference seems to be that their classification approach consisted of two stages. The first stage involved classifying the image as either a tumor or non-tumor. The second stage was executed conditionally: if the image was classified as a tumor, it was then further classified as VT or NC. By using this approach, the authors achieved results similar to ours, despite using a much simpler network with just three convolutional layers that was trained from scratch (as opposed to our transfer-learning approach). This suggests that their hierarchical approach could improve our results as well.

Anisuzzaman et al. (2021) [53] trained several CNNs and reported the best result with VGG19, achieving an overall accuracy of 0.94 using the Adam optimizer and a learning rate of 0.01. Similar to the work of Arunachalam et al. (2019) [35], they followed a holdout strategy and reported their results on a small subset of 230 images. However, our results are markedly different, as we found that VGG19 did not converge on a solution due to the learning rate of 3 × 10^−4^ for the Adam optimizer being too high. When we re-trained VGG19 with a learning rate of 0.01, we still found that it did not converge (data not shown). In addition, the authors reported a very low average *F*1 score with ResNet50, whereas in our experiments, ResNet50 performed well in all cases. Although we tried to follow the authors’ implementation as closely as possible by adding two Fully Connected (FC) layers containing 512 and 1024 neurons to the end of the network, there were still some differences that could have influenced the outcome. Firstly, the authors used Keras applications for importing the VGG19 model, whereas we used the Torchvision implementation contained in PyTorch. Secondly, we used a batch size of 4 for the 512 × 512 input image sizes and 8 for the 256 × 256 image sizes, which differed from the authors’ batch size of 80. Thirdly, the authors downsampled all images to dimensions of 375 × 375 for training and evaluation, while our implementation did not. Lastly, we used a slightly different implementation of the Adam optimizer called AdamW, which corrects the way weight decay is implemented [55]).

Bansal et al. (2022) [59] used a combination of handcrafted (HC) features and Deep Learning (DL) features extracted from the Xception Network to train a Singular Vector Machine (SVM) classifier with a Radial Basis Function (RBF) kernel. They reported an extremely high overall accuracy of 0.995 when tested on a small subset of 219 images, demonstrating a clear advantage when combining features from different methods. In the same work, an overall accuracy of 0.968 with EfficientNetB0 was reported, which is higher compared to our findings. On one hand, this deviation is expected when validating the networks on different sets of samples. We observed folds where our MobileNetV2 achieved an overall accuracy of more than 0.96, but this was not a realistic estimate, as indicated by our cross-validation results. On the other hand, the increased reported accuracy might also be due to a difference in the authors’ approach. They modified the network after training by removing its last FC layer to extract a set of features, which were then filtered using a Binary Arithmetic Optimization Algorithm (BAOA) and classified using an RBF-SVM. Further evaluation using cross-validation with both approaches is required to assess whether this technique can further improve network performance.

### 4.2. Limitations and Future Perspectives

While the present study has yielded valuable insights into the use of deep learning networks for the classification of OS tissue samples, there are several limitations that need to be acknowledged.

Firstly, our study compared the performance of different network architectures using a default set of hyperparameters. While the results showed that smaller networks can outperform larger ones and that MobileNetV2 and EfficientNetB0 were the most effective models, it is possible that the performance of other networks could be improved by optimizing their hyperparameters. Therefore, future research could explore the impact of hyperparameter optimization on the performance of the evaluated networks.

Secondly, the OS dataset contained only a limited number of images that depicted multiple tissue categories, while the majority of images exclusively contained NT, VT, or NC. We observed that after removing ambiguous images depicting both VT and NC, the network’s performance was improved. Thus, we expect the accuracy of the network to be reduced in a scenario where images contain any combination of these tissue categories. Increasing the dataset to contain more ambiguous samples or expanding the annotations to include pixel-level classification for segmentation approaches could result in more reliable results.

Lastly, our study was limited by the size of the OS dataset, which contained images from just four patients that were selected by pathologists based on the diversity of tumor specimens. Different tumor types, stages, or even demographics, may result in unique imaging characteristics. These factors would be likely to impact the generalizability of our findings due to the dataset not being representative of the broader population of OS patients. Thus, larger and more diverse OS datasets are required to confirm the effectiveness of the identified networks and to ensure their generalizability to new datasets and populations. The inclusion of data from multiple centers could help further increase dataset diversity.

## 5. Conclusions

The present study evaluated various deep-learning networks for the classification of osteosarcoma tissue samples. Our results suggested that commonly used deep networks exhibited overfitting, and that smaller networks could outperform larger ones on the present dataset. Specifically, the MobileNetV2 and EfficientNetB0 models led to the most effective overall classification of non-tumors, viable tumors, and necrosis when the original images were downsampled from 1024 × 1024 pixels to 256 × 256. Re-training MobileNetV2 using five-fold cross-validation showed consistent results across all folds, achieving an overall accuracy of 0.91 and mean recalls of 0.95, 0.93, and 0.83 for non-tumors, viable tumors, and necrosis, respectively. Removing images containing both viable tumors and necrosis further improved our results to mean recalls of 0.95, 0.98, and 0.93, and an overall accuracy of 0.96.

These findings suggest that smaller and more efficient networks can be used to improve results on the osteosarcoma dataset without resorting to increasingly bigger and more complex models. Therefore, we recommend that future research focus on evaluating the results of more aggressive regularization techniques, such as pre-training the models on similar but larger datasets, using more creative augmentation techniques, reducing input dimensionality and batch size, and adding dropout [60]. These techniques could achieve greater results in osteosarcoma datasets and serve as an invaluable tool that, when used in conjunction with the expertise and experience of pathologists, could ultimately lead to improved disease outcomes for patients.

## Figures and Tables

**Figure 1 cancers-15-02290-f001:**
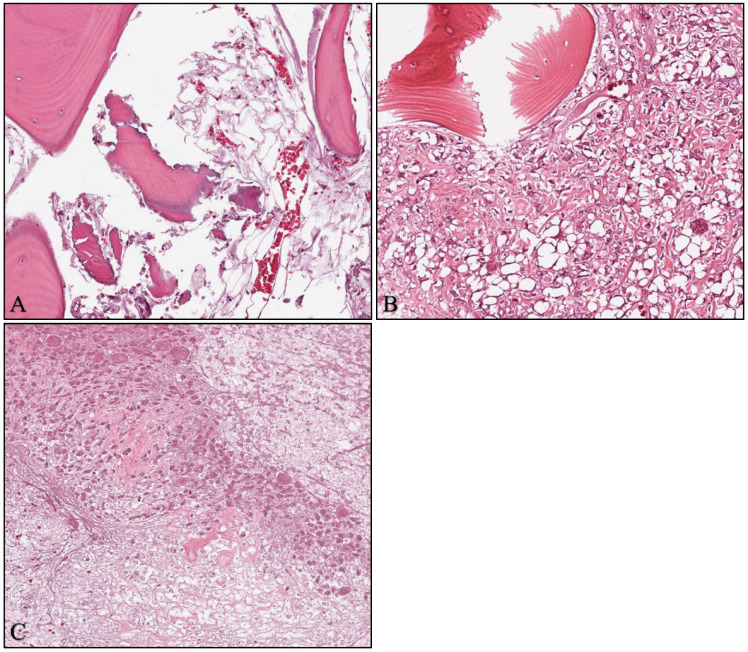
Example images from the Osteosarcoma dataset (magnification × 10). (**A**) Non-tumor; (**B**) viable tumor; (**C**) necrosis.

**Figure 2 cancers-15-02290-f002:**
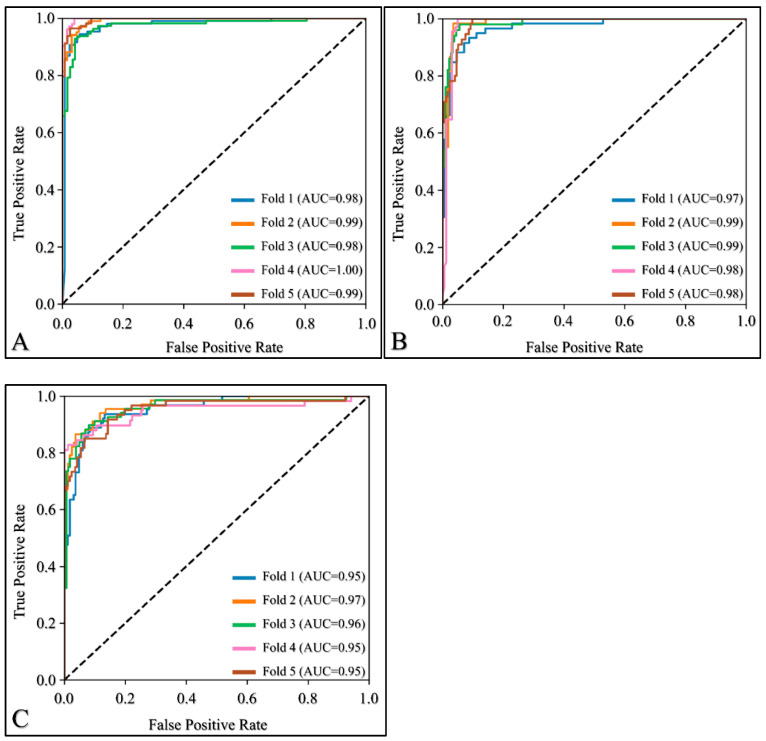
Receiver Operating Characteristic (ROC) curves for (**A**) non-Tumor, (**B**) viable tumor, and (**C**) necrosis.

**Table 1 cancers-15-02290-t001:** The number of parameters for each network variant.

Model	Number of Parameters
EfficientNetB0	4.0 M
EfficientNetB1	6.5 M
EfficientNetB3	11 M
EfficientNetB5	28 M
EfficientNetB7	64 M
MobileNetV2	2.2 M
ResNet18	11 M
ResNet34	21 M
ResNet50	24 M
VGG16	28 M
VGG19	33 M
ViT-B/16	86 M

**Table 2 cancers-15-02290-t002:** *F*1 scores for each network and image size using pre-trained weight initialization.

Network	Image Size	*F*1 Score		
		Non-Tumor	Viable Tumor	Necrosis
EfficientNetB0	1024 × 1024	0.93	0.89	0.84
EfficientNetB0	512 × 512	0.93	0.88	0.83
EfficientNetB0	256 × 256	0.95	0.87	0.85
EfficientNetB1	1024 × 1024	0.93	0.85	0.82
EfficientNetB1	512 × 512	0.94	0.88	0.82
EfficientNetB1	256 × 256	0.95	0.86	0.84
EfficientNetB3	1024 × 1024	0.94	0.89	0.84
EfficientNetB3	512 × 512	0.93	0.86	0.81
EfficientNetB3	256 × 256	0.93	0.87	0.81
EfficientNetB5	896 × 896	0.92	0.89	0.81
EfficientNetB5	512 × 512	0.93	0.87	0.82
EfficientNetB5	256 × 256	0.94	0.84	0.80
EfficientNetB7	512 × 512	0.94	0.88	0.84
EfficientNetB7	256 × 256	0.95	0.87	0.83
MobileNetV2	1024 × 1024	0.82	0.84	0.66
MobileNetV2	512 × 512	0.92	0.85	0.81
MobileNetV2	256 × 256	0.94	0.89	0.85
ResNet18	1024 × 1024	0.83	0.86	0.72
ResNet18	512 × 512	0.92	0.85	0.78
ResNet18	256 × 256	0.92	0.88	0.81
ResNet34	1024 × 1024	0.82	0.87	0.70
ResNet34	512 × 512	0.93	0.92	0.82
ResNet34	256 × 256	0.92	0.92	0.82
ResNet50	896 × 896	0.90	0.89	0.77
ResNet50	512 × 512	0.92	0.88	0.82
ResNet50	256 × 256	0.94	0.89	0.82
VGG16	1024 × 1024	0.63	-	-
VGG16	512 × 512	0.63	-	-
VGG16	256 × 256	0.93	0.89	0.81
VGG19	896 × 896	0.63	-	-
VGG19	512 × 512	0.63	-	-
VGG19	256 × 256	0.63	-	-
ViT-B/16	224 × 224	0.88	0.83	0.72

**Table 3 cancers-15-02290-t003:** Means and standard deviations of the performance metrics of MobileNetV2 over 5 folds.

Metrics	Non-Tumor	Viable Tumor	Necrosis
*F*1 Score	0.95 ± 0.02	0.90 ± 0.04	0.85 ± 0.03
Accuracy	0.95 ± 0.02	0.95 ± 0.02	0.92 ± 0.02
Specificity	0.96 ± 0.03	0.96 ± 0.02	0.96 ± 0.02
Recall	0.95 ± 0.03	0.93 ± 0.05	0.83 ± 0.05
Precision	0.95 ± 0.03	0.88 ± 0.05	0.88 ± 0.05

**Table 4 cancers-15-02290-t004:** Confusion matrix of MobileNetV2 with aggregated results over 5 folds.

	Predicted			
**Actual**		**Non-Tumor**	**Viable Tumor**	**Necrosis**
	**Non-Tumor**	510	7	19
	**Viable Tumor**	3	272	17
	**Necrosis**	24	30	262

**Table 5 cancers-15-02290-t005:** Means and standard deviations of the performance metrics of MobileNetV2 over 5-folds, excluding ambiguous images from the dataset under investigation (ambiguous images were considered those that simultaneously included both NC and VT tissue).

Metrics	Non-Tumor	Viable Tumor	Necrosis
*F*1 Score	0.96 ± 0.03	0.97 ± 0.02	0.93 ± 0.03
Accuracy	0.96 ± 0.03	0.99 ± 0.01	0.97 ± 0.02
Specificity	0.97 ± 0.02	0.99 ± 0.01	0.97 ± 0.04
Recall	0.95 ± 0.06	0.98 ± 0.05	0.93 ± 0.09
Precision	0.97 ± 0.02	0.97 ± 0.03	0.93 ± 0.08

## Data Availability

Not applicable.

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
