# Peer review of "Deep Learning Approaches to Osteosarcoma Diagnosis and Classification: A Comparative Methodological Approach"

_cancers, 2023, doi:10.3390/cancers15082290_

Round 1

Reviewer 1 Report

The authors present a nice use of available image data for a rare pediatric cancer and propose a machine learning tool as most effective in separating class of tissue.

Major comments:

1. the results are presented as an F1 score for each type of ML tool separated only by image size and whether each was pre-trained. While the mathematical mean is presented I am not sure it is relevant to the clinical need to precisely classify each image.

2. The results suggest that there are some outliers in F1 score for normal tissue - in both the untrained and pre-trained tools. how is this explained since there is no information provided describing the different efficientnet types B0- B7

3. were there any efforts made to augment the data set with rotation or inversion to increase the image numbers

4. the manuscript PMID: 30995247 presents an AUC of 0.96 in classifying these images. No effort is made by the authors to compare to these data and reflect on the difference in classification success

5. there is a higher success in the classification of normal tissue across all tools - is this merely a reflection of greater #'s of sample images

Minor comments

1. simple summary - line 14; osteosarcoma is also one of the most aggressive tumors known to present -- who is this being compared to and in what group of patients

2. simple summary line 16 - wording suggests limb salvage surgery is less aggressive - this is untrue and the sentence should be re-worded

3. introduction line 46 - the genetic complexity of osteosarcoma is well described and without references this sentence is poorly worded

4. introduction line 47; OS is the second most common malignancy second to multiple myeloma -- this is incorrect and needs to be reworded

5. introduction line 68 - observer bias needs to be referenced

6. introduction line 69 - automating histological evaluation .... - this sentence is a very big assumption since the results of the project have an 80% precision in some outputs. I might suggest presenting the 'winner' as an ROC curve representing all sensitivity and specificity

7. reference #21 - needs to include authors

Author Response

Reviewer #1

Open Review

(x) I would not like to sign my review report

( ) I would like to sign my review report

Quality of English Language

( ) English very difficult to understand/incomprehensible

( ) Extensive editing of English language and style required

( ) Moderate English changes required
(x) English language and style are fine/minor spell check required

( ) I am not qualified to assess the quality of English in this paper

Yes

Can be improved

Must be improved

Not applicable

Does the introduction provide sufficient background and include all relevant references?

( )

(x)

( )

( )

Are all the cited references relevant to the research?

( )

( )

(x)

( )

Is the research design appropriate?

(x)

( )

( )

( )

Are the methods adequately described?

( )

(x)

( )

( )

Are the results clearly presented?

( )

(x)

( )

( )

Are the conclusions supported by the results?

( )

(x)

( )

( )

Comments and Suggestions for Authors

The authors present a nice use of available image data for a rare pediatric cancer and propose a machine learning tool as most effective in separating class of tissue.

Response: We thank the reviewer for their insightful comments. All comments from the reviewer were taken into account and are addressed distinctively throughout the text.

Major comments:

  1. The results are presented as an F1 score for each type of ML tool separated only by image size and whether each was pre-trained. While the mathematical mean is presented I am not sure it is relevant to the clinical need to precisely classify each image.

Response: The reviewer is correct. We have omitted the mean columns from the tables. Yet, the presented mean which concerned the macro-averaged F1 scores, was used as a metric for the overall model performance. Therefore, we have omitted the respective column, and we have added a Supplementary section explaining what macro-averaging is in a multiclass problem, and also included Supplementary Table S1, with the sorted macro-averaged F1 scores (please refer to Supplementary Table S1, “Results” section p. 7, lines 285-286). In addition, for the follow-up experiment we have added more clinically relevant data, which are presented in Table 3 (new table).

  1. The results suggest that there are some outliers in F1 score for normal tissue - in both the untrained and pre-trained tools. How is this explained since there is no information provided describing the different efficientnet types B0-B7.

Response: We thank the reviewer for the insightful comment. The reviewer is correct. Based on the reviewer’s remark, we have investigated the reason for the outliers. We have observed that our learning rate was too large for our small batch size, and thus the noise in the mini-batches was causing large updates to the weights. This, in turn, resulted in unstable results, that differed significantly between subsequent epochs. To counter this, we introduced cosine annealing learning rate which reduces the learning rate from 3e-4 to 1e-5 over the training course. The text was amended with this information respectively (please refer to p. 5, lines 198-199). After making this change, there were no more unexpected outliers to be observed. In addition, we have amended out text with further details concerning differences between the EfficientNet network variants. As such we have expanded the text in the “Methodology” section (sub-section 2.3.), explaining why there are different variants for each network type (EfficientNets, ResNets, VGGs) and also inserted a new table (Table 1) with the number of parameters for each network variant (please refer to p. 5, lines 175-190).

  1. Were there any efforts made to augment the data set with rotation or inversion to increase the image numbers.

Response: Indeed, data augmentation was used during training but this was not described in the text. We have expanded the text in the “Methodology” section (sub-section 2.3.) with the applied augmentation techniques (please refer to p. 6, lines 228-230).

  1. The manuscript PMID: 30995247 presents an AUC of 0.96 in classifying these images. No effort is made by the authors to compare to these data and reflect on the difference in classification success.

Response: We thank the reviewer for the detailed comment. To address the remark concerning the comparison of our data with previous reports we have added an additional follow-up experiment (please also refer to our response to comment #1), where we re-trained our best performing model with 5-fold cross-validation and compare the results to similar work, including the suggested work. In addition, we have amended our “Discussion” as suggested (please refer to p. 12, lines 416-441).

  1. There is a higher success in the classification of normal tissue across all tools - is this merely a reflection of greater #'s of sample images

Response: We have amended the “Results” section with a respective paragraph, describing the phenomenon (please refer to p. 9, lines 330-342).

Minor comments

  1. Simple summary - line 14; osteosarcoma is also one of the most aggressive tumors known to present -- who is this being compared to and in what group of patients

Response: We have added the respective information in the “Simple Summary”, as well as we have added some epidemiological data in the “Introduction” section (Please refer to p. 1, lines 14-16 and p. 2, lines 16-18).

  1. Simple summary line 16 - wording suggests limb salvage surgery is less aggressive - this is untrue and the sentence should be re-worded.

Response: We thank the reviewer for the detailed comment. We have rephrased the sentence (Please refer to p. 1, lines 16-18).

  1. Introduction line 46 - the genetic complexity of osteosarcoma is well described and without references this sentence is poorly worded.

Response: We thank the reviewer for the detailed comment. We have amended the “Introduction” section significantly, regarding some molecular mechanisms in osteosarcoma (Please refer to p. 1, lines 51-76).

  1. Introduction line 47; OS is the second most common malignancy second to multiple myeloma -- this is incorrect and needs to be reworded.

Response: The reviewer is correct. We have omitted the phrase.

  1. Introduction line 68 - observer bias needs to be referenced.

Response: The reviewer is correct. We have added respective citations (please refer to p. 1, line 96-97).

  1. Introduction line 69 - … automating histological evaluation ... - this sentence is a very big assumption since the results of the project have an 80% precision in some outputs. I might suggest presenting the 'winner' as an ROC curve representing all sensitivity and specificity.

Response: We have down-tuned our expression, as well as we have added a citation concerning the statement (please refer to p. 3, line 98-99). In addition, as suggested by the reviewer, we have amended our “Results” section by performing a ROC analysis of our “winning” network (please refer to p. 10, lines XXX and Figure 2).

  1. Reference #21 - needs to include authors.

Response: We apologize for the inconsistency. We have corrected the reference (please refer to Ref. #43).

Reviewer 2 Report

I have reviewed the manuscript Deep Learning Approaches to Osteosarcoma Diagnosis and Classification: A Comparative Methodological Approach. It used multiple deep learning approaches to compare and classification of HPE of Osteosarcoma using slides HE stained dataset from UT Dallas. The manuscript is very premature. Long introduction, very short results, no cross validation study, no conclusive results to reach the final recommendation. Very add hoc type of approach to reach the conclusion. Results need to be elaborated in depth, cross validation must be performed. Why deep learning methods was chosen, why machine learning based regression, SVM or GMM approach were not attempted. Comparison with other conventional methods must be performed. Statistical test needs to included in results to support and reach conclusion.  Discussion needs to discuss the results of the present study with literature using the same open source dataset, or similar studies. Discussion reads like a student project report rather than a scientific document manuscript. A major revision and thorough rewriting with fresh comparative analysis is required.

Author Response

Reviewer #2

Open Review

(x) I would not like to sign my review report

( ) I would like to sign my review report

Quality of English Language

( ) English very difficult to understand/incomprehensible

( ) Extensive editing of English language and style required

( ) Moderate English changes required

(x) English language and style are fine/minor spell check required

( ) I am not qualified to assess the quality of English in this paper

Yes

Can be improved

Must be improved

Not applicable

Does the introduction provide sufficient background and include all relevant references?

( )

(x)

( )

( )

Are all the cited references relevant to the research?

( )

( )

(x)

( )

Is the research design appropriate?

( )

(x)

( )

( )

Are the methods adequately described?

( )

( )

(x)

( )

Are the results clearly presented?

( )

( )

(x)

( )

Are the conclusions supported by the results?

( )

( )

(x)

( )

Comments and Suggestions for Authors

  1. I have reviewed the manuscript Deep Learning Approaches to Osteosarcoma Diagnosis and Classification: A Comparative Methodological Approach. It used multiple deep learning approaches to compare and classification of HPE of Osteosarcoma using slides HE stained dataset from UT Dallas. The manuscript is very premature. Long introduction, very short results, no cross validation study, no conclusive results to reach the final recommendation. Very ad hoc type of approach to reach the conclusion. Results need to be elaborated in depth, cross validation must be performed.

Response: We thank the reviewer for the fruitful comment. We have significantly re-structured our paper and have expanded our “Results” as suggested. In particular, a follow-up experiment with cross-validation was performed, where additional results were reported in order to support our conclusions (please refer to p. 7, lines 250-264 and p. 9, lines 315-321, 326-329; p. 9, lines 330-342).

  1. Why deep learning methods was chosen. Why machine learning based regression, SVM or GMM approach were not attempted.

Response: We have included a respective paragraph in the “Introduction” section, referring to the use of deep and machine learning methodologies in microscopy for medical purposes. In addition, we have added several citations dealing with this subject (please refer to p. 3, lines 100-114).

  1. Comparison with other conventional methods must be performed. Statistical test needs to be included in results to support and reach conclusion. Discussion needs to discuss the results of the present study with literature using the same open source dataset, or similar studies. Discussion reads like a student project report rather than a scientific document manuscript. A major revision and thorough rewriting with fresh comparative analysis is required.

Response: We have performed additional statistical analyses, we have amended our “Results” section by performing a ROC analysis of our best-performing network (please refer to p. 10, lines XXX and Figure 2). In addition, “Results” were expanded (please refer to p. 7, lines 284-286; p. 8, lines 295-301). To address the remark concerning the comparison of our data with previous reports we have added an additional follow-up experiment (please also refer to our response to comment #1), where we re-trained our best performing model with 5-fold cross-validation and compare the results to similar work (p. 9, lines 315-321; and Tables 3-5). In addition, we have amended our “Discussion” as suggested (please refer to p. 11-12, lines 406-415; p. 12, lines 416-441). A major revision and thorough rewriting has been performed as suggested.

Reviewer 3 Report

The manuscript titled "Deep Learning Approaches to Osteosarcoma Diagnosis and Classification: A Comparative Methodological Approach", highlighted the importance of pre-training and careful selection of network architecture and input image size. The idea of a simple summary is very useful as it introduces a non-biology background person to Osteosarcoma briefly and why this research is important and how machine learning and artificial intelligence approaches can help improve the disease prognosis. I believe it helps the reader to understand the main point of the paper before digging deeper into the paper. The introduction covers mostly the overview of Osteosarcoma and how this research can be important in terms of prognosis. It also explains the usage of machine learning approaches over the past years to this type of dataset and how it is progressing towards better results and how analyzing whole slide images poses a significant challenge.Please find below comments for the manuscript: 

  1. In the methodology section, it is unclear that on what basis the whole slide images are cropped or resized? Also, how and on what basis the image patches are classified?
  2. As they have used an Adam optimizer and because it is widely used in the literature. But it doesn’t have any reference of the literature,  it's recommended to add some references which use Adam optimizer for this kind of data.
  3. It is unclear to me throughout the paper that on what basis the resizing of the images is done as very important information can be missed and while training the data, important features from the images can be overlooked.
  4. The model used in this paper has limitations with the image size which cannot be the case for other types of datasets/images, so this model is very limited.
  5. These authors have applied 5 different platforms for their data and found out that efficient nets performed significantly better than other models whereas none of these models are accurate for a high-resolution image data.
  6. More figures of object detection, segmentation using these models should be displayed for better understanding of the results.

Author Response

Open Review

(x) I would not like to sign my review report

( ) I would like to sign my review report

Quality of English Language

( ) English very difficult to understand/incomprehensible

( ) Extensive editing of English language and style required

( ) Moderate English changes required(x) English language and style are fine/minor spell check required

( ) I am not qualified to assess the quality of English in this paper

Yes

Can be improved

Must be improved

Not applicable

Does the introduction provide sufficient background and include all relevant references?

( )

(x)

( )

( )

Are all the cited references relevant to the research?

( )

(x)

( )

( )

Is the research design appropriate?

( )

( )

(x)

( )

Are the methods adequately described?

( )

( )

(x)

( )

Are the results clearly presented?

( )

( )

(x)

( )

Are the conclusions supported by the results?

( )

( )

(x)

( )

Comments and Suggestions for Authors

The manuscript titled "Deep Learning Approaches to Osteosarcoma Diagnosis and Classification: A Comparative Methodological Approach", highlighted the importance of pre-training and careful selection of network architecture and input image size. The idea of a simple summary is very useful as it introduces a non-biology background person to Osteosarcoma briefly and why this research is important and how machine learning and artificial intelligence approaches can help improve the disease prognosis. I believe it helps the reader to understand the main point of the paper before digging deeper into the paper. The introduction covers mostly the overview of Osteosarcoma and how this research can be important in terms of prognosis. It also explains the usage of machine learning approaches over the past years to this type of dataset and how it is progressing towards better results and how analyzing whole slide images poses a significant challenge.

Response: We thank the reviewer for the fruitful comments.

Please find below comments for the manuscript: 

  1. In the methodology section, it is unclear that on what basis the whole slide images are cropped or resized? Also, how and on what basis the image patches are classified?

Response: We thank the reviewer for the comment, since this is indeed a significant parameter. The selected images, were obtained from a publicly available database, which includes images that are already cropped and consist part of large microscopy slides. Thus, the selected images were taken as provided from the database. We have revised and rephrased the “Dataset” section, including this information (please refer to p. 3, lines 133-134; p. 3-4 lines 143-152).

  1. As they have used an Adam optimizer and because it is widely used in the literature. But it doesn’t have any reference of the literature, it's recommended to add some references which use Adam optimizer for this kind of data.

Response: We have added several references as suggested (please refer to ref. #46, #53, #54, #56, #57 and p. 5, lines 199-202).

  1. It is unclear to me throughout the paper that on what basis the resizing of the images is done as very important information can be missed and while training the data, important features from the images can be overlooked.

Response: It is important to clarify that we did not crop the dataset images during training or validation. Instead, we performed the experiment three times, first on the original full-size images, and then the same, but down-scaled images (Please refer to p. 6 lines 209-221).

  1. The model used in this paper has limitations with the image size which cannot be the case for other types of datasets/images, so this model is very limited.

Response: We thank the reviewer for the insightful comment. Microscopy image analysis, poses several limitations, which are difficult to be overcome due to technical difficulties. Whole slide images can reach sizes of several GBs, which makes it difficult to process, at least in the laboratory setting, with conventional computers (please refer to p. 3, lines 134-140). Yet, this poses also a positive challenge, since it could be useful to be able to classify images of lower size and lower resolution (including parts of images, which have been cropped). For reference reasons, we have added the information of the hardware we used for the present analysis (please refer to p. 5, line 167-169).

  1. These authors have applied 5 different platforms for their data and found out that efficient nets performed significantly better than other models whereas none of these models are accurate for a high-resolution image data. More figures of object detection, segmentation using these models should be displayed for better understanding of the results.

Response: In this present work we do not detect or segment tumors, only classify the whole image, as we do not have pixel-level annotations from the pathologists. As we classify small regions taken from Whole Slide Images (WSIs), we could in theory stitch together the classification of the small patches to perform a low-resolution segmentation. Unfortunately, the current dataset only gives us random patches instead of WSIs.

Round 2

Reviewer 2 Report

Discussion needs more elaborate discussion wrt to results. Line 360 to 405 explains basic differences among different DL algorithms that is very generic. Those can be removed or shortened to large extent. Discussion should focus on comparison of current study results with literature. Authors could also include a limitation section and future work in the discussion section.

Author Response

Reviewer #2

Open Review

Quality of English Language

( ) English very difficult to understand/incomprehensible
( ) Extensive editing of English language and style required
( ) Moderate English changes required
(x) English language and style are fine/minor spell check required
( ) I am not qualified to assess the quality of English in this paper

Yes

Can be improved

Must be improved

Not applicable

Does the introduction provide sufficient background and include all relevant references?

(x)

( )

( )

( )

Are all the cited references relevant to the research?

(x)

( )

( )

( )

Is the research design appropriate?

(x)

( )

( )

( )

Are the methods adequately described?

(x)

( )

( )

( )

Are the results clearly presented?

(x)

( )

( )

( )

Are the conclusions supported by the results?

(x)

( )

( )

( )

Comments and Suggestions for Authors

  1. Discussion needs more elaborate discussion wrt to results.

Response: We thank the reviewer for their comment. We have expanded the discussion section by highlighting differences between the related work and our approach as well as our results (please refer to p. 11, lines 374-382, lines 391-402, and p. 12, lines 409-417).

  1. Line 360 to 405 explains basic differences among different DL algorithms that is very generic. Those can be removed or shortened to large extent. Discussion should focus on comparison of current study results with literature.

Response: The generic network presentation was indeed out of place in the Discussion section. We have removed it completely, as suggested.

  1. Authors could also include a limitation section and future work in the discussion section.

Response: We thank the reviewer for their suggestion. We have added Section 4.2: Limitations and Future Perspectives (please refer to p. 12, lines 420-444).

Reviewer 3 Report

The authors have now mentioned that the images were resized with the help of a pathologist based on tumor heterogeneity. Overall, now explanations are detailed and provide necessary information. The Simple Summary could benefit from some proofreading. The sentences do not flow in some places.

In the discussion section, they discuss how their results do not align with Anisuzzaman et al., it's possible that this paragraph could benefit from a little more explanation into the differences in the methods that could lead to the difference in results

There are few grammatical errors.

Author Response

Open Review

(x) I would not like to sign my review report

( ) I would like to sign my review report

Quality of English Language

( ) English very difficult to understand/incomprehensible

( ) Extensive editing of English language and style required

(x) Moderate English changes required

( ) English language and style are fine/minor spell check required

( ) I am not qualified to assess the quality of English in this paper

Yes

Can be improved

Must be improved

Not applicable

Does the introduction provide sufficient background and include all relevant references?

(x)

( )

( )

( )

Are all the cited references relevant to the research?

(x)

( )

( )

( )

Is the research design appropriate?

( )

(x)

( )

( )

Are the methods adequately described?

(x)

( )

( )

( )

Are the results clearly presented?

( )

(x)

( )

( )

Are the conclusions supported by the results?

(x)

( )

( )

( )

Comments and Suggestions for Authors

The authors have now mentioned that the images were resized with the help of a pathologist based on tumor heterogeneity. Overall, now explanations are detailed and provide necessary information.

Response:

Please find below comments for the manuscript: 

  1. The Simple Summary could benefit from some proofreading. The sentences do not flow in some places.

Response: We thank the reviewer for the comment, as the grammatical errors went unnoticed. We have revised the Simple Summary text.

  1. In the discussion section, they discuss how their results do not align with Anisuzzaman et al., it's possible that this paragraph could benefit from a little more explanation into the differences in the methods that could lead to the difference in results.

Response: We thank the reviewer for the suggestion. We now clearly indicate all identified differences between the two approaches (please refer to p. 11, lines 391-402).

  1. There are a few grammatical errors.

Response: We have proof-read our text and made several minor changes throughout.
